# SNN-LPCG: Spiking Neural Networks with Local Plasticity Context Gating for Lifelong Learning

## Abstract

Humans learn multiple tasks in succession with minimal mutual interference, through the context gating mechanism in the prefrontal cortex (PFC). The brain-inspired models of spiking neural networks (SNNs) have drawn massive attention for their energy efficiency and biological plausibility. To overcome catastrophic forgetting when learning multiple tasks in sequence, current SNNs for lifelong learning focus on memory reserving or regularization-based modification, while ignoring the cognitive control behavior in the brain. Inspired by biological context-dependent gating mechanisms found in PFC, we propose SNNs with context gating trained by the local plasticity rule (SNN-LPCG) for lifelong learning. The iterative training between global and local plasticity for task units is designed to strengthen the connections between task neurons and hidden neurons and preserve the multi-task relevant information. The experiments show that the proposed model is effective in maintaining the past learning experience and has better task-selectivity than other methods during lifelong learning. Our results provide new insights that the SNN-LPCG model is able to extend context gating with good scalability on different SNN architectures. Thus, our models have good potential for parallel implementation on neuromorphic hardware for real-life tasks.

## 1 Introduction

The human brain can encode and preserve new memories continuously without disrupting previously acquired memories, that is, the ability of lifelong learning. It is believed that the primate prefrontal cortex (PFC) associates the cognitive control of selecting context-appropriate tasks and executing with minimal interference by gating task-irrelevant input dimensions Miller et al. (2001); Flesch et al. (2022). However, it remains unclear how synaptic plasticity induced by past experiences of old tasks is maintained when receiving new experiences of new tasks. Inspired by spike computation, network organization, and plasticity learning in the brain, Spiking neural networks (SNNs) have exclusive advantages in energy efficiency and biological plausibility computation Xu et al. (2022). However, affected by catastrophic forgetting, most of the current SNNs agents often need to learn samples of all tasks in a randomly mixed way to learn new tasks Zhang et al. (2018); Xu et al. (2021), which is quite different from human multi-task lifelong learning. Hence, we aim to explore the possible solution to implement the lifelong learning of SNNs with context gating induced by synaptic plasticity, to model human "cognitive control" behavior while minimizing the intervention of artificial inference.

The most essential points in lifelong learning are the capability to perform competitively on the previous task after learning on the subsequently observed dataset. However, the vanilla SNNs trained with surrogate gradient fail to learn multiple tasks in sequence without suffering from catastrophic forgetting, in which the performance decreases sharply on previously encountered tasks after learning new ones. To overcome that problem, the SNNs for lifelong learning have been proposed such as memory reserving Mozafari et al. (2019), regularization-based method Skatchkovsky et al. (2022), or reward-based method Allred & Roy (2020). But these models ignore the intrinsic property of cognitive control behavior when proceeding with lifelong learning in the brain.

In this paper, our goal is to implement the context gating by the local synaptic plasticity to solve the lifelong learning for SNNs and explore whether SNNs could help understand and implement lifelong learning like the human brain. To this end, we develop single-spike and multi-spike SNN-LPCG and train them with human "cognitive control" experimental samples and analyze the influence of local plasticity on the performance of the model and the possible reason behind it. The contributions are as follows:

- The SNNs with context gating learned by local plasticity (SNN-LPCG) framework is proposed for lifelong learning, which can alleviate catastrophic forgetting to a certain extent via local plasticity induced context gating instead of manual weight updating fixation or the samples had been learned.

- The single-spike and multi-spike SNN-LPCG models are implemented with the local spike-timing-dependent plasticity (STDP) and Oja learning rules. The iterative training between global backpropagation and local plasticity is designed to strengthen the connections between task neurons and hidden neurons to preserve the multi-task relevant information.

- The experiments are conducted to analyze the reason for the effectiveness of local plasticity context gating during SNNs' lifelong learning. Besides, the performance of the proposed model is compared with the characteristics of human behavior in the cognitive experiments. The results suggest that the above two show a considerate degree of consistency. Moreover, the proposed model shows prior biological plausibility of task neurons' selectivity in neural dynamics.

## 2 RELATED WORKS

Lifelong learning aims to remember previously trained old tasks and acquire new knowledge by training new tasks. The direct trained SNNs models suffer from the catastrophic forgetting problem when proceeding with multiple-task continuous training. Different approaches have been developed to overcome catastrophic forgetting.

The regularization method preserves synaptic connections that are deemed to be essential to consolidate previous knowledge. Considering the synaptic noise and Langevin dynamics, the regularization-based method for SNNs in Antonov et al. (2022) determines the importance of synaptic weights by stochastic Langevin dynamics with local STDP instead of gradient. Based on Bayesian, each synaptic weight is represented by parameters that quantify the current epistemic uncertainty resulting from prior knowledge and observed data. The proposed online rules update the distribution parameters in a streaming fashion as data are observed Skatchkovsky et al. (2022).

Memory replay indicates that maintaining representative samples or anchors with the key features of previous tasks as episodic memory promotes lifelong learning. The SNNs trained by R-STDP eliminate catastrophic forgetting with small episodic memory in Mozafari et al. (2019). The cortical connections between excitatory neurons are plastic and regulated by STDP. The results show that sleep is able to prevent catastrophic forgetting through spontaneous reactivation (replay) of both old and new memory traces.

Other lifelong learning methods of SNNs are proposed based on the observed neuroscience mechanisms. Inspired by the dopamine signals in mammalian brains, the controlled forgetting network model is introduced to address the stability-plasticity dilemma with local STDP learning to control the forgetting process and reduce the accuracy degradation on new tasks Allred & Roy (2020). Especially, context gating has been suggested to be effective in supporting lifelong learning. For instance, inspired by that neuron in PFC codes for specific tasks and exert top-down control to prioritize context-appropriate stimuli and action, the context-dependent gating signal along with the input is introduced to guarantee that only sparse, mostly non-overlapping patterns of neurons are active for each task in Masse et al. (2018). In this method, between $80\%$ and $86.7\%$ of hidden units are gated on the MNIST dataset for peaked classification accuracy. The optimal gated unit numbers depend on the network complexity and task numbers and need to be tuned by parameter validation.

However, since the neurons in PFC code for specific tasks and exert top-down control to prioritize context-appropriate stimuli and actions, these methods ignore the dynamic task selectivity of neuron populations during the cognitive control process.

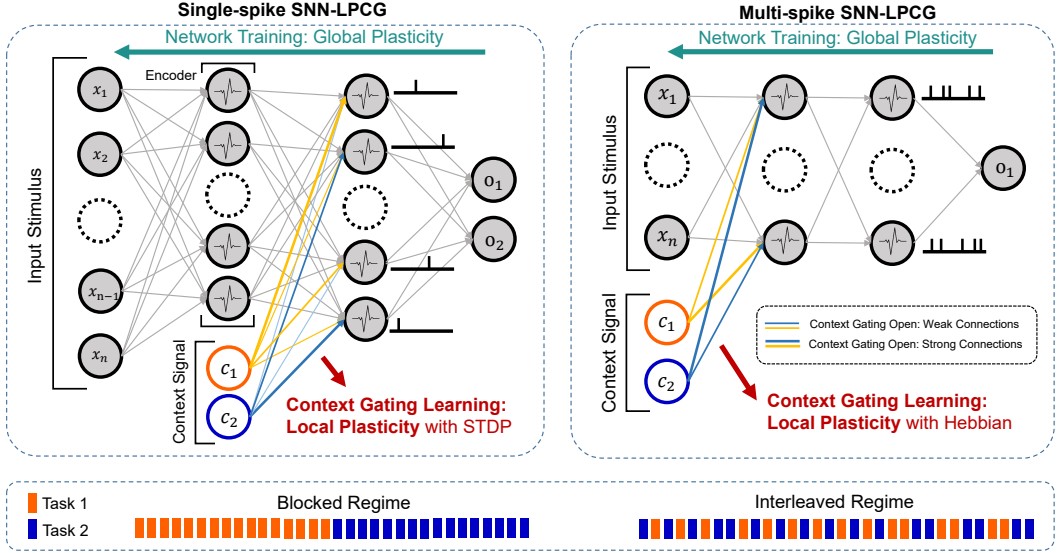

Figure 1: The framework of the network construction of the single-spike SNN-LPCG and multi-spike SNN-LPCG. (Top) Two different kinds of learning regimes: blocked and interleaved regimes. Human learn better under a blocked curriculum when tasks are not similar. (left bottom) The network structure of single-spike SNN-LPCG. The first encoding layer codes the input stimuli into spike trains and avoids weak signal ignorance. The additional context signals of tasks are introduced into the second layer. The iterative training between the global and local plasticity proceeds during the training process, in which all the synaptic weights are updated by the global backpropagation rule, and the weights of the context signal are updated by the local plasticity rule. (right bottom) The network structure of multi-spike SNN-LPCG. The network is a simple feed-forward MLP with two hidden layers with IF spiking neurons that receive the pixels information of pictures together with the one-hot contextual signal as input. And the connections between the contextual cue and the first layer will be successively updated by the Hebbian learning rule. (right bottom) shows the blocked and interleaved distribution of the two tasks' samples.

## 3 METHODS

In this paper, we aim to explore the possibility of applying local plasticity, which can enhance the connection between context input and corresponding activated dendritic segments, to make SNNs achieve the effect of lifelong continued learning. As mentioned earlier, it is generally believed that human "cognitive control" is closely related to PFC. There is evidence that PFC can prioritize context-appropriate stimuli and actions through top-down control. In some past continued learning work, such as Duncker et al. (2020) and Zeng et al. (2019), the gated parameters are mostly updated by human coding or fixation, which lacks biological plausibility. Based on these considerations, we choose the spiking model as the basic framework for the design and construction of the target networks. The data from relevant cognitive experiments followed by Flesch et al. (2018) (Supplementary material for details) is used to analyze and evaluate the modeling ability and performance of the proposed model.

### 3.1 NETWORK ARCHITECTURE

In the above experiment, participants need to classify the input stimuli according to the given task rules. We model this behavior as three-layer MLP spiking-neural networks, in which the main part of the network is used to receive pictures stimuli and output results, while the contextual signals are encoded by one-hot labels and then input into the network so as to affect the network's judgment. And we use local plasticity to enforce the connection between the context inputs and relevant characters.

Based on different spike coding methods, We develop SNN-LPCG based on two different kinds of SNNs, multi-spike SNNs trained with the surrogate gradient and single-spike SNNs with direct backpropagation. Both SNNs use stochastic gradient descent (SGD) algorithms to train on blocked or interleaved data. Once a supervised training step is finished, the network will follow a step (some trials more than one) of local plastic learning.

### 3.1.1 SINGLE-SPIKE SNN-LPCG

We employ the basic SNNs models in Mostafa (2018), where each neuron is permitted to fire only once. The non-Leaky IF neuron model is utilized for its simplicity and efficiency. The membrane potential of each postsynaptic neuron $j$ is obtained after integrating the contributions of all the weighted presynaptic neurons of $N_I$:

$$V_j(t) = \sum_{i=1}^{N_I} g(t - t_i) w_{ij} (1 - exp(-(t - t_i))),\tag{1}$$

where $g(x)$ denotes the Heaviside function, which equals zero for negative arguments of $x$, otherwise equals to be one for positive ones. $t_i$ indicates the concrete firing time of presynaptic neuron $i$. Thus, the presynaptic neuron $i$ would transmit the corresponding postsynaptic potential to neuron $j$ only if it satisfies $t_i < t$; otherwise, that potential vanishes. The postsynaptic neuron $j$ emits a spike when its membrane potential crosses the threshold $V_{th}$ which is set to 1. Due to each neuron being permitted to fire only once, the spike firing times $t_j$ is transformed to z-domain by $exp(t_j) \rightarrow z_j$ to simplify implementation. Meanwhile, the casual set $C_j = \{i : t_i < t_j\}$ is defined as the collections of the presynaptic spikes that determine the time point at which the postsynaptic neuron fires the first spike. Since then, the firing times of each neuron can be computed sequentially over the feedforward process through the above formulations.

In the experimental process, we find that directly using the single-spike structure in Mostafa (2018) is easy to make the network ignore the relatively weak signal (that is, the darker pixels) in the source visual stimulus. Therefore, we apply an MLP structure at the first layer, which is used to extract the features of the source image. The hidden layer then encodes the extracted features (the value is limited by implementing a sigmoid function) together with the contextual signal according to their signal strength for forward propagation.

Besides, to strengthen the difference in STDP's influence on different synapses, we try to encode the stronger contextual signal with the same intensity as the strongest signals on the hidden layer during Hebbian updating. We also adjust the number of weight updating of global plasticity and local plasticity in a round in order to coordinate the strength between them.

**Context gates by STDP.** The local plasticity of STDP is a reasonably well-established physiological mechanism of activity-drive synaptic regulation based on the local adjacent neurons. And it has been observed extensively in vitro for more than a decade Feldman (2000), and it is believed to play an important role in neural activity Masquelier et al. (2009).

$$\Delta W_{ij}^{stdp} = \begin{cases} A_+ exp(\frac{t_j - t_i}{\tau^+}), & if\ t_j > t_i, \\ A_- exp(\frac{t_i - t_j}{\tau^-}), & if\ t_j < t_i, \end{cases}\tag{2}$$

where $A_+, A_-$ and $\tau^+, \tau^-$ indicate the magnitudes and time constants, respectively. Thus, the $w_{ij}$ is updated by $w_{ij} = w_{ij} + \lambda_{stdp} * \Delta W_{ij}^{stdp}$.

**Loss function** Since the modeled behavior is a binary classification task, we use a simplified cross-entropy to calculate the loss. Besides, we multiply the loss value by the reward value as the participants would receive the numerical reward after their decision in cognitive experiments.

$$L_{single} = -r \times \frac{exp(-o_0)}{exp(-o_0) + exp(-o_1)},\tag{3}$$

where $r$ is the corresponding numerical rewards of the trials, $o_0, o_1$ is the spike latency of the last layers.

### 3.1.2 Multi-spike SNN-LPCG

The global training of the multi-spike SNN-LPCG model is implemented based on spatiotemporal backpropagation (STBP) algorithm in Wu et al. (2018). The iterative integrate-and-fire (IF) neuron is employed to express neuronal dynamics both in the spatial domain and temporal domain. According to the integrated presynaptic neuron input $I(t)$ and the membrane potential at $t-1$, the membrane potential of postsynaptic neuron $j$ at $t$ can be computed as:

$$u_i(t+1, n) = u_i(t, n) + x_i(t+1, n) + b_i, \tag{4}$$

$$x_i^{t+1,n} = \sum_{j=1}^{l(n-1)} w_{ij}^n o_j^{t+1,n-1}, \tag{5}$$

$$o_i^{t+1,n} = g(u_i^{t+1,n} - V_{th}), \tag{6}$$

The Heaviside function of $g(x)$ is to guarantee that the spike fires once the membrane potential cross over the threshold $V_{th}$.

Then, the surrogate gradient of $g_{(}x)$ is approximated based on the derivative of spike activities of $g'(x) = \frac{\alpha}{2(1+(\frac{\pi}{2}\alpha x)^2)}$, to solve the non-differential problem of discrete spikes firing behavior, where $\alpha$ is set to be 2.0. Based on the above formulation and the gradient computing in Wu et al. (2018), the update of synaptic weights can be obtained by the gradient descent rules.

During experiments, we find that the effect of STDP learning rules in the multi-spike model was very meaningless. We infer that this result might be due to the fact that the positive contextual signal is powerful and is encoded to a relatively high frequency in the multi-spike network. Thus, the local plasticity learned by STDP becomes unstable, which makes STDP rules difficult to assist the network to distinguish functional neurons. Therefore, we don't apply STDP rule to multi-spike model as the local plasticity rule.

**Oja rules.** Inspired by Flesch et al. (2022), we use Oja's rule Oja (1982), which can regularize the weight based on Hebbian learning, to constrain the parameters.

$$w = w + \eta_{hebb} y(x - wy), \tag{7}$$

where $\eta_{hebb}$ is the learning rate of the Hebbian update, $x$ is the inputs and $y$ is the linear hidden units output connected to the inputs $x$ via weight matrix $w$.

**Sluggish neurons.** According to the ordinary cognition and the conclusion from experiments of human volunteers Flesch et al. (2018), Humans usually can learn tasks better after blocked training than interleaved training, while the artificial neural network is usually on the contrary. Inspired by Flesch et al. (2022), we apply the "sluggish" neurons to model the human's learning bias under interleaved training. That is, when learning in interleaved mode, the relation between the context cues and stimuli of the former samples may interfere with the judgment of the current sample. The sluggish neurons will carry the information from previous samples.

By simply using Hebbian learning rules and SNNs, we can not only achieve basic cognitive control in the blocked learning regime but also have higher task-specific neural codes because of the detailed neural dynamics compared with artificial neural networks.

## 4 Results

In this section, we will conduct several experiments to validate the effectiveness and the biological plausibility of our method. The grid search is employed to find suitable parameters for each model. The final results are obtained by running the network over three times and taking the average results for comparison.

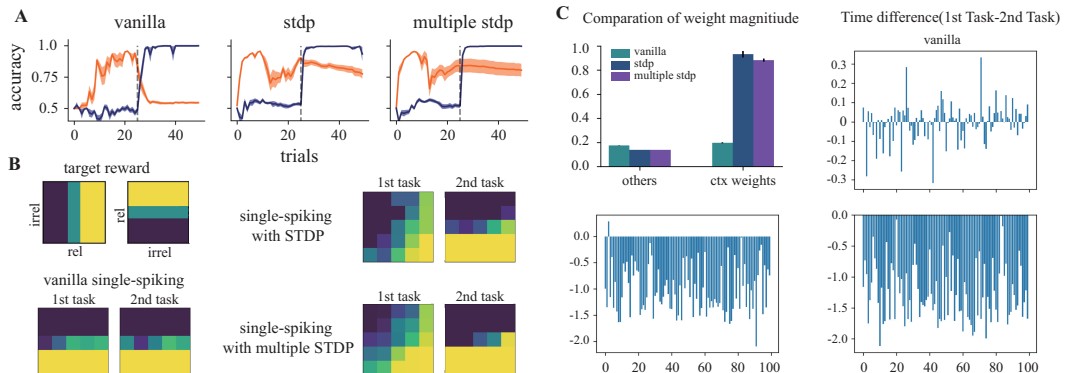

Figure 2: (A) The accuracy curve of single-spike SNN-LPCG. (left) Vanilla network of single-spike SNN. (mid) vanilla network with STDP learning rules. (right) vanilla network with multiple STDP update. (B) (left top) The target choice of the given data. (other) Plotting the choice of the trained network with (right top) vanilla setting, (left bottom) STDP learning rules, and (right bottom) multiple STDP updating. (C) (left top) The average absolute values of weights related to the context signal input and other weights in the hidden layer of three networks. (other) The time difference of spiking time of hidden layer neurons of a trained network under two different contextual signals. They are vanilla networks (right top), networks with STDP learning rules (left bottom), and networks with multiple STDP updating (right bottom).

### 4.1 THE EFFECTIVENESS OF SINGLE-SPIKE SNN-LPCG LOCAL PLASTICITY WITH STDP

We first explore how STDP may take effect during context-dependent task learning. As illustrated in Fig. 2 (A), as expected, we note that single-spike SNN is also affected by catastrophic forgetting, and the learning performance of the first task almost immediately drops down. In contrast, when applying STDP learning rule, the decline rate of the first task is restrained to a very slow level, and the performance of the second task can still rapidly increase until convergence to perfect performance. The accuracy of the first task finally stays at about ∼75%. In addition, the model with multiple STDP updating seems to have stronger resistance to catastrophic forgetting while its results are relatively unstable.

However, although the memory decline decelerates under the effect of STDP, the accuracy curve under the first task becomes even more unstable than the vanilla network. Then we compare the average absolute values of the weights related to the context signal and other weights in the hidden layer of the three models. As shown in Fig. 2 (C) (left top), under the influence of STDP learning rule, the absolute value of context-related weight becomes very large. Besides, after training under the blocked data, the time difference of hidden layer neurons' spike between two tasks is very close (mostly lower than 0.05ms, the largest is only ∼0.3ms) and evenly distributed, which makes the choice matrix of the first task almost replaced by that of the second one. In contrast, the hidden layer neurons of the network using STDP learning rules always fire spike under the first task earlier than under the second task, which makes the network distinguish the two independent tasks to a certain extent.

Based on these phenomena, we infer that the relatively large weight magnitudes caused by STDP learning rules enable the network to structure some cognition of the past tasks to a certain extent so that it can make correct choices toward relatively strong stimuli after being trained under the second task. This can also be reflected in Fig. 2 (B) (down). The choice matrix of the first task smoothly changes along the areas where the choices are consistent under the two tasks. The large magnitude of the weight might also be the main reason for the instability of the accuracy curve in the first task.

Though having a certain effect, the stability and the final performance of the model using STDP learning rules in context-dependent tasks are still not satisfactory. The neural dynamics of the brain are constructed by a large group of neural plasticity mechanisms Churchland et al. (2012) Ahrens et al. (2012) Golub et al. (2018). The exploration and discovery of single-spike SNN-LPCG might provide some ideas for related research fields. Based on the problems in the single-spike model, we

further explore multi-spike SNN-LPCG, which can implement a more stable and better model of human cognitive control behavior, and has better biological plausibility.

Figure 3: (A) (upper) the network accuracy change curve of the vanilla blocked training (left), applying OWM learning algorithm on the latter two connections (mid ) and our method (right). (bottom left) applying OWM learning algorithm on all the connections. (bottom right) the standard reward value of the whole task and the choice of the trained network in two dimensions under the vanilla blocked training network (left two), the network with OWM applied in the latter two layers (mid two), and the network trained under our method (right two). (B) The impact of the sluggish neurons in context-dependent decision making task. (left top) The accuracy of neural networks trained on interleaved data (left) or blocked data (right) with different levels of "sluggishness". (right top) the proportion of units in the hidden layer which are task selective (obtained by linear regression). (left bottom) Linear coefficients were obtained from a regression of the output against the models shown in the midden. (mid bottom) The two models are used to model the possible choice of the human. The factorized model means the two features are learned separately while the linear model means the two features are learned chaotically and ignore the context cue. (right bottom) The network output for different levels of sluggishness.

## 4.2 COMPARED MULTI-SPIKE SNN-LPCG WITH HUMAN MODIFICATION

To validate the effectiveness of multi-spike SNN-LPCG, we compare it to other lifelong learning methods under the same setting, such as the regularization-based lifelong learning method of the orthogonal weights modification (OWM) learning algorithm in Zeng et al. (2019). As shown in the Fig. 3 (A), in contrast to the vanilla multi-spike model, the lifelong learning models with the OWM algorithm and our multi-spike LPCG perform better on both two tasks, because these models retain the context information learning before a certain extent. Besides, the third network's performance is much better than the second one while the second model applies more strict constraints on the network connections. It may reveal that excessive human constraint sometimes will impact the network's ability to retain memory.

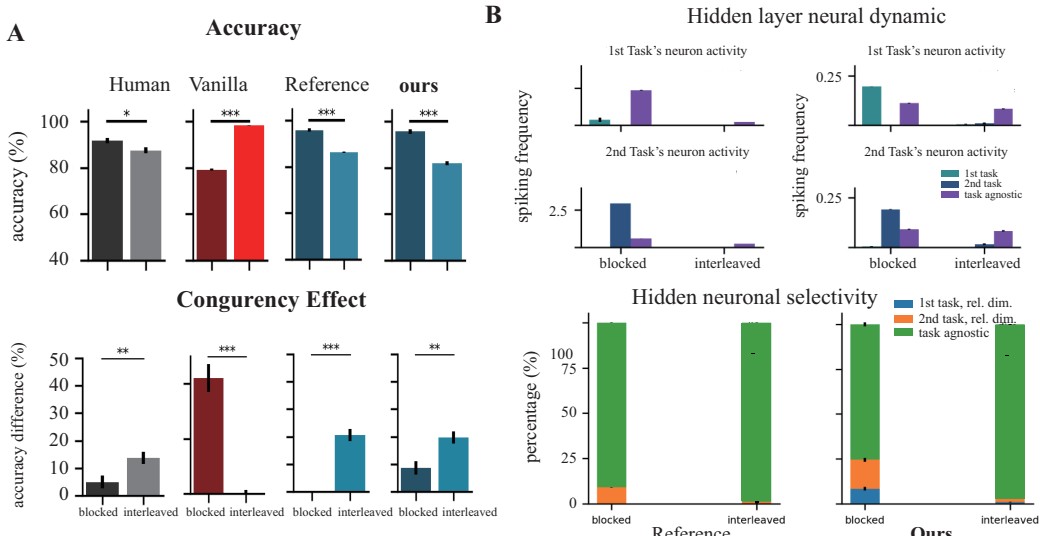

Figure 4: (A) (top) Test accuracy of human data and three network data: (left) human data, (mid left) vanilla network, (mid right) Flesh et al.'s model, (right) our model. (bottom) The comparison of congruency effect. (B) (top) Proportion of task selective units under blocked and interleaved training of the reference network (left) and our networks (right). (bottom) The average hidden layer activity of two networks, and we use spiking frequency for SNNs while using activation value for artificial neural network because the framework is different. (B) (top) Proportion of task selective units under blocked and interleaved training of the reference network (left) and our networks (right). (bottom) The average hidden layer activity of two networks, and we use spiking frequency for SNNs while using activation value for artificial neural network because the framework is different.

Moreover, comparing the accuracy curve of the OWM learning algorithm, our multi-spike LPCG method can retain the learned tasks to a greater extent under the same circumstances. Especially, the accuracy curve of our method has almost stopped falling while the curve of the OWM algorithm still shows a downward trend in the last several trials. And on the bottom figure in Fig. 3 (A) , we can see the detail of the choices made in two dimensions. The vanilla network trained under blocked training almost ignores the task signal, and the OWM model and our SNN-LPCG model can keep the choices learned from the first task. Meanwhile, the network trained with our method can keep more details near the category boundary of the first tasks.

### 4.3 MODEL THE COGNITIVE CONTROL BEHAVIOR OF HUMAN

We conduct experiments to evaluate the effectiveness of the sluggish neurons. In Fig. 3 (B), when $\alpha = 0$ it means the network is the same as the vanilla network. Combining the results of Fig. 3 (B) (top) and 3 (B) (left bottom), we can conclude that as the level of sluggishness increases the performance of the network decreases. At the level of neural dynamic, the proportion of the task-selective hidden layer units, which are active only under specific contextual cues, is also reduced. Meanwhile, the choice matrix of the network under interleaved training becomes more and more like the linear model in Fig. 3 (B) (mid bottom) as the sluggishness values become larger. On the other hand, the same statistical data of the network trained in blocked data with local plasticity hardly be affected by sluggishness. Therefore, the "sluggish" neurons are reasonable assumptions. Meanwhile, though the best accuracy obtained by the trained network under interleaved data (without sluggish) is relatively high, the largest proportion is still lower than that of the trained network under blocked data.

Then after reasonably modeling the human behavior under interleaved data, we can use our method to try modeling the behavior of the human participants in Flesch et al. (2018). And to access the rationality and the biological plausibility of our method, we compare the validation performance and neural dynamics after blocked or interleaved training among the experimental data, the vanilla spiking network, the model proposed in Flesch et al. (2022), and our method.

Firstly, as shown in Fig. 4 (A), we can see that participants trained under blocked data perform better than those trained under interleaved data while the vanilla network shows the opposite phenomenon. The model of Flesch et al. (2022) and our model both appear the similarity to human behavior. And participants' behaviors show the congruency effect during the training period, which means that they perform better on the congruent trail between two tasks than the incongruent trials. This effect is stronger when training in the interleaved data, as shown in Fig. 4 (A) (bottom). We can see that the model of Flesch et al. (2022) and our model both reveal such a congruency effect while the vanilla network can't reproduce such an effect because of the catastrophic forgetting.

Then when fitting the psychometric functions (sigmoid) to the choice made by human participants and the other three models (in supplementary Fig.1 ), we can see that our model and the reference model have relatively steeper slopes for the irrelevant dimension under interleaved training than blocked training as shown in human performance. But the vanilla network obtains steeper slopes for the irrelevant dimension under blocked training because the catastrophic forgetting makes the SGD network more easily affected by the irrelevant feature. A similar result also can be seen in supplementary Fig.1 (bottom). We fit the factorised model and the linear model (shown in Fig. 3 (B) (mid bottom)) to the choice of human and three networks. Human participants learned the clear boundary of two tasks/dimensions under blocked data and learned a relatively vague boundary under interleaved data. The reference network and the network with our method can model such characteristics in human behavior while the performance of the vanilla neural network is almost the opposite.

In summary, these tests can demonstrate that our method can overcome the catastrophic forgetting of the neural network. And our framework can effectively model human cognitive behavior mode and achieve the similar effect to the framework proposed by Flesch et al. (2022). In addition to the modeling of human behavior patterns, we also compared the neuron dynamics in our framework with theirs, to prove the biological plausibility of our framework.

Though the framework in Flesch et al. (2022) exhibits great neuron selectivity in the abstract context-dependent task samples (Gaussian "blobs") and reaches a relatively great accuracy in trees picture (in Fig. 4 (A) (top)). It shows poor task-selectivity in their network when dealing with trees picture.

When under blocked training, their network only has a very small proportion of task-selectivity units in the hidden layer ($\sim 10\%$) while in our network a considerable proportion of units is selective to a specific task ($\sim 25\%$). As shown in Fig. 4 (B), the number of the units which are only active to the first task even drops to $\sim 1\%$. Besides, we plot the average neural activity of task-selective neurons in Fig. 4 (B) (bottom). In our network, the task-selective neurons have a higher spiking frequency than the task-agnostic neurons when facing the corresponding task. The reference network exhibits poor performance on the 1st task-selective neuron's activity. This phenomenon indicates that our model has better biological plausibility and neural dynamics during human cognitive control.

## 5 CONCLUSION

Affected by catastrophic forgetting, SNNs agents often need to learn samples of all tasks in a randomly mixed way to learn new tasks, which is quite different from human multi-task lifelong learning. Therefore, in this paper, we have extensively explored the role of local synaptic plasticity in context gating of SNNs, which is more biologically plausible, so as to model human "cognitive control" behavior while minimizing the intervention of artificial inference. Taken together, based on the gating in the PFC, we studied and explored whether the known local plasticity rules can promote SNNs to possess the effect of context gating. The single-spike and multi-spike SNNs with context gating trained by local plasticity were implemented. The models we proposed have revealed better effectiveness in maintaining the past learning experience. Besides, we fitted our network to the previously published human behavior data and found that our method can reproduce the human behavior data in both blocked and interleaved data. Moreover, our model indicates prior neural dynamics of task neuron selectivity compared with the previous work.

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
