# SNN-LPCG: Spiking Neural Networks with Local Plasticity Context Gating for Lifelong Learning

## 1 Data source

The modeled behavior data comes from the context-dependent categorisation tasks conducted by Flesh et al. on human participants Flesch et al. (2018). In that experiment, participants needed to distinguish relevant information from two independent features (the density of branches, called "branchiness", and density of leaves, called "leafiness") to decide whether to accept/reject the given pictures (various types of tree) according to two different contextual cue (types of the garden, north or south) without prior knowledge. The participants would then receive a numerical reward according to their choice, which was positively correlated with the level of leafiness in the north garden and the level of branchiness in the south garden, after their decision. Participants were either trained successively under blocked samples or interleaved samples. Both groups were evaluated on an interleaved test block without feedback.

In this paper, we used the reduces image samples and contextual signals from that experiment to train the networks, and compared the performance of the networks with the characteristics of human participants.

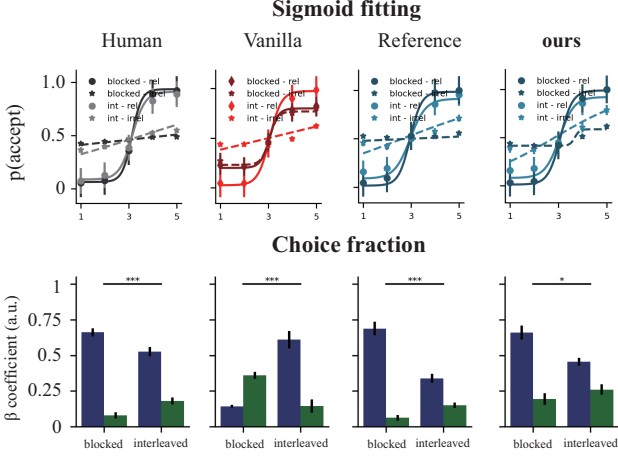

Figure 1: (top) Sigmoidal fits of choice of four conditions mentioned in Figure 4(A). (bottom) Linear fitting of factorized and linear model to four conditions.

## References

Timo Flesch, Jan Balaguer, Ronald Dekker, Hamed Nili, and Christopher Summerfield. Comparing continual task learning in minds and machines. *Proceedings of the National Academy of Sciences*, 115(44):E10313–E10322, 2018.