# OpenReview forum: "SNN-LPCG: Spiking Neural Networks with Local Plasticity Context Gating for Lifelong Learning"
_ICLR.cc/2024/Conference — ICLR 2024 Conference Withdrawn Submission_

### Official Review · Reviewer_mMb4 · 2023-10-29

**Soundness:** 3 good
**Presentation:** 3 good
**Contribution:** 2 fair
**Rating:** 5
**Confidence:** 3

**Summary:**

This paper explores the problem of catastrophic forgetting in SNN-based lifelong learning and proposes a biological plausible method to overcome it. The authors develop SNN-LPCG, inspired by human cognitive mechanism that promotes memory replay, on two different kinds of SNNs, multi-spike SNNs trained with the surrogate gradient and single-spike SNNs with direct backpropagation, alternatively updated with supervised and local plastic learning steps that induce context gating instead of manual weight updating fixation. The experiments show that their method alleviates catastrophic forgetting and has better task selectivity and neural dynamics than previous frameworks in modeling human cognitive behavior. In particular, they apply “sluggish” neuron in modeling the human’s learning bias that favors blocked rather than interleaved data, revealed importance of task neuron’s selectivity in neural dynamics.

**Strengths:**

1.	In this paper, the authors propose a novel continual learning method combines gradient backpropagation and brain-inspired local updating rules to build a SNN-based model for continual learning, and captures crucial attributes of human learning in experimental data, which could potentially contribute to establish more biologically plausible computational cognitive models for human learning paradigm.
2.	The authors also discuss the distinct behavior of memorizing past training data in SNN-LPCG in contrast to traditional methods, allowing for mitigated forgetting in a way that is similar to human brain, which is an intriguing idea that may be beneficial for some learning settings with constrained data availability.

**Weaknesses:**

1.	The experiments focus on a somewhat simplified data setting. It would be convincing if more experimental results on more continual learning benchmarks are available.
2.	Lack of satisfactory explanation of how “sluggish” neurons benefit continual learning despite sub-optimal final performance. While the paper demonstrates the effectiveness of sluggish neurons in capturing human cognitive behavioral patterns in context-dependent decision-making tasks, it does not provide a clear interpretation of how these neurons affect the overall performance of the model.
3.	Distinct local plastic learning rules are applied in single- and multi-spike SNN-LPCG, respectively. Such comparison would be somewhat unfair if no further justifications are provided.

**Questions:**

1.	What is the motivation of introducing “sluggish” neurons beyond enhancing interpretability in terms of cognitive mechanisms? Further refinement is required to discuss its role in continual learning.
2.	Why different local plastic rules are used in single- and multi-spike SNN-LPCG?

---

### Official Review · Reviewer_eBXQ · 2023-10-30

**Soundness:** 3 good
**Presentation:** 2 fair
**Contribution:** 3 good
**Rating:** 6
**Confidence:** 5

**Summary:**

This paper proposes SNNs with context gating trained by the local plasticity rule (SNN-LPCG), to overcome catastrophic forgetting when learning multiple tasks in sequence and achieve lifelong learning. The iterative training between global and local plasticity rule is applied to strengthen the connections between task neurons and hidden neurons and thus to preserve the multiple task-relevant information in SNNs. The experiments are conducted to evaluate the effectiveness of local plasticity context gating when solving the catastrophic forgetting of lifelong learning using SNNs. It shows that the proposed SNNs model has a prior biological plausibility of task neurons’ selectivity in neural dynamics.

**Strengths:**

1. The paper is easy to follow.

2. The proposed SNN-LPCG model applied SNNs to model human behavior in cognitive experiments and found that the SNNs perform better about the task neurons’ selectivity in neural dynamics than ANNs. It is a novel exploration to exploit the potential of SNNs in lifelong learning cognitive tasks. Meanwhile, the application of the local plasticity of SNNs, especially the spike firing time-based STDP rule, is suitable for implementing the context gating for SNNs, this attempt gives a unique preview for the latter research of the SNNs on lifelong learning tasks.

3. The experiments are sufficient to demonstrate both the biological plausibility and computational efficiency of the proposed model.

**Weaknesses:**

1. The related works lack references to continual learning with SNNs in the recent year.

2. The presentation of the parameter setting is not sufficiently clear and detailed. Please clarify it.

3. This paper lacks a crucial illustration that highlights the significance of cognitive control in the context of solving lifelong learning tasks. It would greatly enhance the paper's clarity to include a compelling explanation of why cognitive control is necessary in addressing these tasks.

**Questions:**

1. What is the parameter setting about the learning rate of the Oja rule in Equation 7?

2. The effect of the spike-based local plasticity when modeling the cognitive control in SNN-LPCG could be explained further.

3. It seems that the mechanism of local plasticity adapts to other structures of SNNs, such as the convolutional and recurrent spiking neural networks, how the proposed method can be implemented in the above models? Is there any limitation for your model when applied to other SNNs with different structures?

---

### Official Review · Reviewer_HidL · 2023-10-31

**Soundness:** 2 fair
**Presentation:** 1 poor
**Contribution:** 1 poor
**Rating:** 3
**Confidence:** 2

**Summary:**

The author(s) explore means for lifelong learning, specifically the alleviation of catastrophic forgetting, in spiking neural networks. In particular, the authors utilise local plasticity rules and other biologically motivated features in order to learn multiple tasks and capture human experimental data during blocked/interleaved training.

**Strengths:**

- the authors are able to capture human like accuracy and a congruency effect in blocked/interleaved trials, though it is unclear to me how much this builds upon the existing reference model (see below)
- I do personally believe local plasticity can be very useful as a means for lifelong learning - to that end I believe there is motivation for the research undergone by authors. However, I clarify that again it is very unclear to me if substantial progress is made (see below)

**Weaknesses:**

- I had substantial difficulty in reading and understanding this work. I am sure this is in part due to a lack of expertise in SNNs but I do believe that overall the presentation is poor. See questions/clarifications below
- It is still not clear to me how context-dependent gating - a fundemental feature of this work - is actually implemented in this work. So far as I can tell, the local plasticity rules (STDP in eq 2, Oja in eq 7) provide the implementation for context-dependent gating (one subsection is called 'context gates by STDP'). How these rules are context dependent, or provide context gating, I don't understand. I believe Figure 1 provides the best illustration of how this is employed, and there is some text context input is used in 3.1, but I am desperate for a more formal description/equations (where is the context signal in eq 2/7? how do these mediate synapses in the SNN?). Moreover, if these are indeed the means for context-dependent gating, the fact that the single-spike and multi-spike models employ different rules (STDP + Oja respectively) makes it very difficult to identify the key computation which enables lifelong learning. Or should I understand that many types of local plasticity rules are (perhaps in their own way) beneficial for learning multiple tasks?
- Relatedly, it is not clear to me why the proposed model performs better than vanilla SNNs for remembering previously learned tasks. Can the authors suggests what latent factors are alleviating catastrophic forgetting? Most figures are dedicated to accuracy, selectivity and choice metrics, which are certainly helpful to see, but I would like to better understand why/how these local plasticity rules seem to preserve important weights and maintain performance. The authors do consider weight magnitude of STDP vs vanilla networks, but I don't see larger weights by themselves are beneficial for catastrophic forgettnig (and as the authors describe, these are prone to instability)
- It is unclear to me how the proposed model extends upon the Flesch et al. (2022) model. Many of their results seem very similar and capture human data well. What is the Flesch model lacking that this model possesses? The authors argue that the "reference network exhibits poor performance on the 1st task-selective neuron’s activity. This phenomenon indicates that our model has better biological plausibility and neural dynamics during human cognitive control." I don't understand this. What does poor performance on activity mean? Both models seem to learn equally well (if anything the reference model a tiny bit better). Is low selectivity less biologically plausible?
- A more detailed description of the task is necessary somewhere in the main text.

**Questions:**

- In the abstract the authors describe an "iterative training between global and local plasticity for task units". In the methods/results I failed to find a description/clarification of this training regime.
- The lack of brackets for many of the references makes the paper very awkward to read. Please fix.
- the phrase 'cognitive control' is used multiple times throughout the paper, but what it actually means is not clear to me. This should be more formally described.
- often the article 'the' is used unnecessarily. For example, in the introduction: 'the SNNs', 'the context gating', 'the local synaptic plasticity'
- Figure 1 caption. I believe (left/right bottom) should read (left/right top), and (right bottom) should just read (bottom)
- The authors claim that other approaches to continual learning (Dunckner 2020, Zeng 2019) which depend on fixation lack biological plausibility, why is this?
- The authors in their methods mention 'the construction of the target networks' in section 3 - what is the target network? Does it provide labels for supervised learning?
- In section 3.1 it's written that "we use local plasticity to enforce the connection between the context inputs and relevant characters." What do 'relevant characters' mean?
- In eq 2 is capital W the same as small w? i.e. is W_{ij} = w_{ij}?
- below eq 2 in the weight update for stdp there is a lambda term, what is this?
- In 3.1.2 first paragraph I believe 't - 1' should be 't' and 't' should be 't + 1'
- Equations 4 - 6, what is n? Why is u used here to representation membrane potential but V in equation 1?
- Equation 7 requires better notation/explanation. Are these entire populations or single neurons/synapses? It seems like the whole population but previously only updates for single neurons/synapses have been given. In the equation we have x - wy - should this not be y - wx?
- Section 4 first paragraph: by 'parameters' this is strictly 'hyperparamters' I presume (and the model weights being trained with local plasticity/backpropagation of error)? and 'three times' means 'three random initialisations of the network'?
- In Fig 2, what is 'multiple stdp' and how is it different from normal stdp
- In Fig 2A a legend is needed for task A/B. In Fig C x,y labels are required
- In section 4.2, is the 'third network's performance' with respect to the XdG model? What are the 'strict constraints on the network connections' this model possesses?
- The effect of sluggish neurons is shown (Fig 3B), but I'm not sure if there is any novelty to these results. Is this effect only shown for the vanilla networks, or was the effect of sluggishness also considered for the proposed model with local plasticity rules?
- Section 4.3, a description (if only brief) of the Flesch et al. model should be given.

---

### Official Review · Reviewer_dZw5 · 2023-11-02

**Soundness:** 2 fair
**Presentation:** 1 poor
**Contribution:** 2 fair
**Rating:** 1
**Confidence:** 4

**Summary:**

Artificial neural networks suffer from difficulties in learning multiple tasks, a problem that, in the extreme, can result in catastrophically “forgetting” how to perform a previously-learned task as the weights are updated while learning a new one.  In contrast, humans do not generally have such problems.  However, the performance of both can be greatly impacted by the learning paradigm.  ANNs tend to perform better when tasks are interleaved together so that they learn both quasi-simultaneously compared to sequentially learning one task and then another, while humans exhibit the converse. The authors modify spiking neural networks to allow contextual information to gate local Hebbian plasticity in a manner meant to simulate such mechanisms in prefrontal cortex. Moreover, the authors allow for so-called “sluggish” neurons that allow for an amount of memory in the interleaved task to carry over across transitions. The method results in more task-selective neurons than a different method they compare to, and overall seems to result in less catastrophic forgetting compared to some other methods.

**Strengths:**

The authors provide some background on the problem of catastrophic forgetting in the context of ANNs and humans, and provide what may be a novel approach to the problem with some promising results.

**Weaknesses:**

1. Overall the manuscript is not of publication quality, as it is not well-written and contains many errors, oversights, typos, figure organization problems, and a general lack of clarity. Some examples of these problems are listed below. While there are some promising results, I believe the manuscript needs a major rewrite and careful re-reading before resubmitting to another conference or journal.

2. Sluggish neurons appear to be core to the authors’ model, yet I was unable to find them defined.  In the model, $\alpha$—which is used to define the sluggishness in the figures as ($1-\alpha$)—refers to the numerator of the surrogate gradient function $g(x)$ and is said to be set to 2.0 (note, $g(\cdot)$ is doubly defined, as it refers to the Heaviside function in Eq. 1).

3. The authors observe that their method results in more task-selective neurons compared to the “Reference” Flesch et al. 2022 model, and that this “indicates that our model has better biological plausibility and neural dynamics during human cognitive control.”  But as no biological evidence from humans is provided for task-selective neurons, nor their relative proportion, no biologically-based adjudication can be made between the 2 models from this criterion.
   - Moreover, Fig. 3B top right is only brought up with the discussion of Fig. 4b.  These panels should appear together and in order of reference of the main text

4. Given that the authors are aiming for greater biological plausibility in the context of catastrophic forgetting, the model neuron used should minimally be the standard LIF (leaky integrate-and-fire) neuron, not one that does not have a leak term.

5. The main Flesch et al. 2018 experiment should be briefly described in the Introduction, not in the Supplemental Material, so that the reader can better understand the motivation behind the model and the analysis thereof without having to download a separate file
   - For example, while “contextual signals” is mentioned and later modeled, the reader can’t know what that means with respect to this experiment without an initial explanation of the experimental setup and task(s)
   - Similarly, the fact that subjects received a reward is suddenly brought up when introducing the loss function rather than at the beginning

6. The authors compare their model to different models at different times.  E.g., they compare their model to OWL / XdG in Fig. 3a, and then to the “Reference” model from Flesh et al. 2022 in Figs 3 and Supp. Fig. 1.

7. The multi-spike model is not well described, with many variables shown but not described.  E.g., $I(t)$ is mentioned but does not appear in the model, it’s unclear what $n$ is, and while several of the variables will be easily inferred, each should be well-described after appearing in the model

8. Generally, there seem to be too few trials to make firm conclusions about the models’ final accuracies, as many have downward trends at the end of the trials that disallow any final conclusions.  Examples include Fig. 2a, STDP and Multiple STDP and Fig. 3a OWM and XdG.

9. There are a number of models, analyses, and other important aspects that are not well described.  For example:
   - XdG, plotted in Fig. 3a, is not described or referenced (it comes from Masse et al. 2018) in either the main text or the figure caption
   - The “Reference” Flesch et al. 2022 model is not described.
   - The choice matrices are not explained at all, and the linear model is only cursorily described (“the two features are learned chaotically and ignore the context cue” in Fig. 3 caption).
   - Statistical significance tests in Fig. 3a are not described at all
   - Error bars/shading not described at all

10. Fig. 3 needs to be split into 2 separate figures, and more panels labeled (A/B/C, i/ii/ii) to avoid top/bottom etc. labels.  Currently the caption is difficult to read and associate the text to the panel under discussion.

11. I could not find any description or explanation in the main text for Fig. 3b bottom (Choice Fraction).

*Minor*

12. The second “contribution” seems to simply provide some detail to how the framework proposed in contribution 1 is implemented rather than to describe a contribution to the field.

13. Fig. 1: Top and bottom mixed up between figure and caption

14. It’s indicated that the spike times are mapped onto the z-domain, but no explanation of the z-domain is given, nor is it obvious as to why exponentiating the spike times would simplify the implementation

15. Some of the main text for Fig. 2A reads “The accuracy of the first task finally stays at about ∼75%.”  Yet there appears to be a clear downward trend, rather than plateauing, at the end.  It seems this comment should be taken out without further trials.

16. Fig. 2a needs to have a legend.

**Questions:**

I’m not sure I understand the rationale for choosing this loss function.  Is there an intuitive reason why the ratio of the exponentials of the spike latencies (dividing through by $\exp(-o_0)$) of the final layers should contribute to the learning process?